# Retrospective Evaluation of Method of Treatment, Laboratory Findings, and Concurrent Diseases in Dairy Cattle Diagnosed with Left Displacement of the Abomasum during Time of Hospitalization

**DOI:** 10.3390/ani12131649

**Published:** 2022-06-27

**Authors:** Theresa Tschoner, Yury Zablotski, Melanie Feist

**Affiliations:** Clinic for Ruminants with Ambulatory and Herd Health Services at the Centre for Clinical Veterinary Medicine, LMU Munich, Sonnenstrasse 16, 85764 Oberschleissheim, Germany; y.zablotski@med.vetmed.uni-muenchen.de (Y.Z.); melanie.feist@lmu.de (M.F.)

**Keywords:** abomasal rolling, abomasopexy, concurrent diseases, dairy cattle, Dirksen, Janowitz, right flank laparotomy

## Abstract

**Simple Summary:**

Left displacement of the abomasum (LDA) is a common disease in high-producing dairy cattle, resulting in direct and indirect costs, discomfort for the cattle, and death if not treated. The objectives of this retrospective study were to assess the effect of treatment on recovery from LDA during time of hospitalization, to investigate the influence of concurrent diseases on the recovery, and to identify prognostic indicators in laboratory findings. Neither the number of concurrent diseases nor the method of surgery had any influence on the outcome (death or recovery). The most common concurrent disease was metritis/endometritis (38.4% of cows). Conservative treatment (abomasal rolling) was successful in 92.8% of cows, with a recurrence rate of 56.7%. Neither oral nor analgesic therapy had any influence on the recurrence of LDA following abomasal rolling during hospitalization. In cows undergoing surgery, endoscopic abomasopexy as described by Janowitz and right flank laparotomy were performed most often (40.8% and 40.2%, respectively). A significantly (*p* < 0.01) higher number of cows showed the outcome “recovery” compared with “death”. The results of this study indicate that the outcome after surgery for LDA under clinical conditions does not depend on the method of surgery, and that concurrent diseases are often diagnosed in cattle with LDA. Conservative treatment has a high recurrence rate.

**Abstract:**

Left displacement of the abomasum (LDA) is a disease often diagnosed in high-producing dairy cattle, resulting in direct and in indirect costs for the farmer, and discomfort and death for the cows. For the present retrospective study, the aims were to assess the effect of treatment on recovery during the time of hospitalization of the cows, to investigate the influence of concurrent diseases on the recovery, and to identify prognostic indicators in laboratory findings. Metritis/endometritis (38.4% of cows) was the concurrent disease diagnosed most often. Conservative treatment (abomasal rolling) was performed successfully in 92.8% of cows; the recurrence rate was 56.7%. Neither treatment with an oral drench nor treatment with analgesics had any influence on the recurrence of LDA following abomasal rolling during hospitalization. Endoscopic abomasopexy as described by Janowitz was performed more often than right flank laparotomy (40.8% and 40.2%, respectively). A significantly (*p* < 0.01) higher number of cows had the outcome “recovery” compared with “death”. The results of this study show that the outcome after surgery for LDA under clinical conditions does not depend on the method of surgery. Moreover, cows with LDA often suffer from concurrent diseases. If conservative treatment is decided on, farmers should be informed that there is a high recurrence rate, and other treatment options should be discussed.

## 1. Introduction

In high-producing dairy cattle, left displacement of the abomasum (LDA) is a common disease [1,2], causing massive economic losses due to treatment costs, reduced milk yield, and a higher risk of culling [3]. About 85% to 91% of cases of LDA occur in the first 6 weeks after calving [4,5]. In female cattle, LDA is more common than dilation/torsion of the abomasum [6]. The etiology is multifactorial and has been discussed elsewhere [5,6,7]. Risk factors, such as prepartum nutrition and management, the composition of feedstuff, negative energy balance, hypocalcemia, gas production, and hypomotility of the abomasum can result in the development of LDA [2,4,5,8]. Incidence rates have been described as ranging from 3% to 5% [5] and 0.35% to 4.4% [4] in North American herds and 1.2% to 2.6% in Holstein herds in Germany [4].

Prognosis depends slightly on the chosen treatment [4] and early diagnosis. Treatment options adopt either a conservative approach, closed or open surgical procedures [4,6,7], or minimally invasive techniques [4]. Conservative treatment consists in casting the animal on its right side for several minutes [4,6] and rolling the animal slowly and clockwise in a 180 degree arc [3,7]. For Blind Tack/Toggle Pin, which is a closed procedure, the abomasum is sutured to the body percutaneously after rolling the cow [4,7]. Open surgical procedures used for the correction of LDA are right flank omentopexy, right paramedian abomasopexy, left paralumbar abomasopexy, or left flank laparoscopy [3].

Studies on the outcome after surgical techniques for the correction of LDA [3,6,9,10,11] and comparisons of the different surgeries [3,12] have been published. However, an evaluation of surgical techniques used for the correction of LDA and the outcome for cattle during time of hospitalization in Bavaria, Germany, has not been published yet.

The objective of the present study was to retrospectively (1) investigate the effect of surgical or conservative treatment on recovery from LDA (outcome) during time of hospitalization, (2) assess the influence of concurrent diseases on the recovery, and (3) identify prognostic indicators from an analysis of laboratory blood parameters based on the outcome in cattle referred to the Clinic for Ruminants with Ambulatory and Herd Health Services of the Ludwig Maximilian University (LMU) Munich, in Bavaria, Germany, by field veterinarians or farmers with (suspected) diagnosis of LDA, and confirmed diagnosis of LDA at the clinic.

## 2. Materials and Methods

### 2.1. Animals

The medical records of 718 cows (≥2 years of age and ≥first lactation) which had been admitted to the Clinic for Ruminants with Ambulatory and Herd Health Services, LMU Munich, between the 1st of January 2009 and the 31st of December 2019 were analyzed retrospectively. The Clinic for Ruminants serves an area of farms around Munich and in Bavaria, and also admits cows from Baden Wurttemberg and Austria. A mean of 756 cattle patients are treated in the clinics as either in-patients or ambulatory patients per year. The clinic is one of five university clinics for veterinary medicine in Germany, offering education and training for students, as well as European or national specialization programs for veterinarians. The clinic is an animal species clinic. Cattle which were included in the present data set were referred to the clinic either by veterinarians working in field practices or by farmers, based on the (suspected) diagnosed of LDA. All cattle admitted to the clinic with a diagnosis of LDA were identified using logbooks and the clinic’s electronical database and were included in the data research for the study if they had a diagnosis of LDA at the clinic. The data of 252 cows which were diagnosed with LDA and were submitted to a diagnostic examination of claw lesions were also included in another publication about the evaluation of claw lesions, inflammatory markers, and outcome after abomasal rolling [13].

### 2.2. Diagnosis of Left Displacement of the Abomasum

Cows were referred to the Clinic for Ruminants with Ambulatory and Herd Health Services for a second opinion/clinical examination by veterinarians working in the field, or farmers, with a suspected diagnosis of LDA, or unclear diagnosis. At the clinic, the cows were submitted to a full clinical examination [14] by a veterinarian, including auscultation and percussion of the left flank, as well as a rectal examination, an analysis of the chloride concentration of the rumen fluid after passing a stomach tube into the rumen, a laboratory blood analysis, and in some cases which were unclear, an ultrasonographic examination to diagnose the LDA or make another diagnosis.

### 2.3. Review of Medical Records

The information that was retrieved from the medical records included signalment of the cows (age, sex, breed), calving date, and days post-partum.

### 2.4. Conservative or Surgical Therapy and Outcome of Therapy

Treatment of LDA was evaluated. Treatment was either performed fully conservatively (“abomasal rolling”), by surgery, or both. The decision of which treatment a cow should receive (conservative or surgical) was made by the veterinarian performing the admission examination. Conservative treatment was often performed in cows with lameness and/or obvious claw pathologies during diagnostic claw trimming [13], in animals with severe deterioration of the general condition (often including severe ketosis or severe changes in the electrolyte metabolism), and during emergency service hours due to lack of time of the veterinarian on duty to perform surgery. Due to the retrospective nature of the present study, explanatory statements about how the decision was made can no longer be obtained.

Abomasal rolling was defined as casting the animal on a hydraulic tilt table in lateral recumbency. The abomasum was not tacked, toggled, or fixated in any other way. For cows treated with abomasal rolling, treatment with analgesia and/or oral therapy (oral drench) following abomasal rolling was recorded (Table 1).

From 2009 to 2019, surgeries were performed by a total of 20 veterinarians working at the clinic. Due to the high number of veterinarians over the period of 10 years, and the variable combination and number of veterinarians performing the surgeries, the influence of the surgeon was not included in the statistical model. Surgical procedures were grouped as follows: “right flank laparotomy” in the standing animal, including omentopexy (fixation of the greater omentum at the right body wall to hold the abomasum in a near anatomically correct position with sutures), pyloropexy (fixation of the abomasum by passing sutures trough the antrum of the pylorus), or omentopexy as described by Dirksen (fixation of the greater omentum at the right body wall by means of a button [15]); “left flank omentopexy” in the standing animal (fixation of the greater curvature of the abomasum at the ventral body wall), or “endoscopic abomasopexy” as described by Janowitz (placement of a toggle pin within the lumen of the abomasum and deflation under laparoscopic guidance in the standing animal, and fixation of the greater curvature of the abomasum at the ventral body wall with the animal in dorsal and lateral recumbency [16]). A detailed description of the different surgical methods can be found in [17]. All surgeries were performed with local anesthesia of the surgical site (either by nerve block or infiltration anesthesia).

For the present paper, the outcome of therapy was defined either as discharge from the clinic (=recovery) or as non-survival due to euthanasia or death (=death) at the clinic during or after surgery. Moreover, the duration of the stay at the clinic and the recurrence rate of LDA after conservative treatment were considered as the outcome. Due to the long period of the retrospective data collection, the farmers were not contacted about the survival period after discharge from the clinic.

### 2.5. Review of Concurrent Diseases

The diagnosis of concurrent diseases ((puerperal) metritis, mastitis, septicemia, subclinical ketosis (BHB from 1.2 to 2.99 mmol/L [18]), clinical ketosis (BHB > 2.99 mmol/L [18], hypocalcemia (ionized Ca < 1 mmol/L as defined by the reference ranges used by the Clinic for Ruminants with Ambulatory and Herd Health Services), claw diseases, septicemia) during hospitalization at the Clinic for Ruminants with Ambulatory and Herd Health Services was recorded. Due to the retrospective nature of the study, the diagnoses of retained fetal membranes, metritis, and endometritis were summarized as one disease; all forms of mastitis are indicated as “mastitis”, and claw pathologies are given as “claw disease”. The manner of diagnosis of concurrent diseases is presented in Table 2:

### 2.6. Review of Biochemical Parameters of Blood Analysis

The results of biochemical analysis from the day of admission exam (packed cell volume (PCV), leucocyte and thrombocyte count, pH, base excess, anion gap, glucose, L-lactate, protein, aspartate aminotransferase (AST), glutamate dehydrogenase (GLDH), gamma glutamyl transferase (GGT), sodium (Na), potassium (K), ionized Ca (Ca), phosphorus (P), magnesium (Mg), betahydroxybutyrate (BHB), and glutaraldehyde test) were evaluated.

### 2.7. Statistical Analysis

The relationships between year and surgery type, between outcome and surgery type, and between recurrence and number of concurrent diseases were studied via Pearson’s chi-square of independence. The pairwise comparisons between subgroups of the categorical variables were conducted with Fisher’s tests due to a (often) low (*n* < 5) number of observations per category. Benjamin–Hochberg correction of *p*-values was applied for multiple Fisher’s tests. The distribution of numeric variable—duration at clinic—was tested using the Shapiro–Wilk normality test. Due to a non-normal distribution, a Kruskal–Wallis test was performed to compare duration at clinic among surgery types. Pairwise Dunn tests for comparisons between particular surgery types with Benjamin–Hochberg correction of *p*-values followed the Kruskal–Wallis test. The influence of age and breed on the outcome was studied via univariate logistic regressions. The influence of laboratory parameters on the outcome was first studied via univariate logistic regressions. Laboratory parameters with a *p* < 0.2 were then considered for the multivariate model. Backwards stepwise elimination via Akaike’s information criterion with an inclusion criterion of *p* < 0.05 was then applied in order to (1) control for confounding factors and (2) reduce the number of variables to only potentially influential ones, while (3) at the same time maximizing model quality. Variance inflation factors (VIF) in the final multivariate model were used to check the assumption of multicollinearity. Statistical significance was considered at *p* ≤ 0.05. All statistical analyses were performed using the R version 4.0.3 (2020-10-10, R Foundation for Statistical Computing, Vienna, Austria). 

## 3. Results

### 3.1. Animals

The medical records of 718 cows with a diagnosis of LDA were analyzed. Out of these, 672 cows were included in the statistical model. A total of 13 cows were excluded because the methods of surgery were not indicated correctly and thus were not analyzable, and 33 cows were excluded due to missing charts or data.

All cows included in this study were female. The breeds were German Simmental (66.4%, *n* = 446), Holstein Frisian (24.3%, *n* = 163), Red Frisian (0.9%, *n* = 6), Brown Swiss (3.0%, *n* = 20), Crossbred (4.2%, *n* = 28), and Other (1.2%, *n* = 8). The cows were 5.1 ± 1.8 (1.8 to 16.1) years old; age was not given in one animal. Days post-partum were given for 573 cows and ranged from 0 to 127 (13.8 ± 13.1) days.

### 3.2. Conservative Treatment of LDA

In three cows, information about abomasal rolling (“yes” or “no”) was not given; these cows were not included in the statistical model. The results of conservative treatment are presented in Table 3. After abomasal rolling, LDA was solved in a significantly (*p* < 0.01) higher number of cows, compared with “not solved”. Animals with LDA “not solved” were hospitalized at the clinic for 9.1 ± 5.5 days (1 to 21 days). Surgery was performed in 71.8% (*n* = 150) of the cows after abomasal rolling.

Following abomasal rolling, 45% (*n* = 94) cows were treated with an oral drench, and 87.1% (*n* = 182) with one (79.0%, *n* = 165) or two (8.1%, *n* = 17) analgesic drugs, respectively. An oral drench in combination with either one or two analgesic drugs was administered to 76 and 11 cows, respectively.

Neither oral therapy (drench, *p* = 0.12) nor the administered number of analgesic drugs (0, 1, or 2, *p* = 0.44) had an influence on relapse after rolling. A list of all components of analgesic and oral therapy is provided in Appendix A. The numbers of concurrent diseases in the cows submitted to abomasal rolling were zero in 11.0% (*n* = 23), one in 35.4% (*n* = 74), two in 32.5% (*n* = 68), three in 15.8% (*n* = 33), and four in 5.3% (*n* = 11) of the cows, respectively.

### 3.3. Surgical Treatment of LDA

A total of 81.4% (*n* = 547) of the cows were submitted to surgery. Endoscopic abomasopexy as described by Janowitz [16] was performed in 40.8% (*n* = 274) of the cows; in one animal, endoscopic abomasopexy was aborted and right flank laparotomy was performed.

Right flank laparotomy was performed in 40.2% (*n* = 270) of the cows; omentopexy as described by Dirksen [15] was performed in 76.7% (*n* = 207) of the cows, with the abomasum not being displaced in 55 of these cows. Omentopexy was performed in 8.5% (*n* = 23) of the cows, with the abomasum not being displaced in three cows; pyloropexy was conducted in 7.4% (*n* = 20) of the cows, with the abomasum not being displaced in two cows. In one animal (0.4%), the abomasum was dislocated to the right during surgery, and in one animal (0.4%), the abomasum could not be fixated due to diagnosis of a peritonitis. The method of right flank laparotomy was not indicated in 0.7% (*n* = 2) of the cows.

Left flank laparotomy was performed in 0.5% (*n* = 3) of the cows. No surgery was conducted in 18.6% (*n* = 125) of the cows, with spontaneous remission in 22.4% (*n* = 28) of these cows.

The length of stay of the cows at the clinic and the outcome according to surgery are presented in Figure 1. The method of surgery did not influence the number of days spent at the clinic.

### 3.4. Outcome

A significantly higher (*p* < 0.01) number of cows had the outcome “recovery” (80.2%, *n* = 539) compared with “death” (19.8%, *n* = 133). Neither breed (*p* = 0.051) nor age (*p* = 0.16) had any influence on the outcome. Age in years and days post-partum were 5.2 ± 1.8 years (2 to 16.1) and 14.0 ± 13.8 days (1 to 127) for “recovery” and 4.9 ± 1.9 (2 to 11.9) and 13.2 ± 9.8 (0 to 66) for “death”.

In cows treated with right flank laparoscopy or endoscopic abomasopexy, a significantly higher number of cows recovered, compared with cows being euthanized (*p* < 0.01, respectively). In cows which were not submitted to surgery, a significantly higher number (*p* < 0.01) were euthanized or died, compared with the number that recovered (Table 4).

### 3.5. Concurrent Diseases

The concurrent diseases evaluated for this study were metritis/endometritis, mastitis, septicemia, subclinical and clinical ketosis, hypocalcemia, and claw diseases. Laboratory findings for ionized calcium were missing in eight cows; therefore, these were excluded from the statistical model, resulting in *n* = 664 cows overall and *n* = 294 cows for cows treated with abomasal rolling. A total of 82.4% (*n* = 547) of the cows were diagnosed with at least one concurrent disease. The distribution of concurrent diseases is presented in Table 5. The number of concurrent diseases was zero for 17.6% (*n* = 117), one for 35.1% (*n* = 233), two for 30.0% (*n* = 199), three for 12.0% (*n* = 80), four for 4.8% (*n* = 32), five for 0.3% (*n* = 2), and six for 0.2% (*n* = 1) of the cows. The number of concurrent diseases did not influence the survival rate (recovery or death, *p* = 0.88), but there was a trend for concurrent disease to influence the spent at the clinics. A higher number of concurrent diseases decreased the number of days spent at the clinics (*p* = 0.08) due to the higher risk of being euthanized. The influence of the type of concurrent disease on recovery or death (survival rate) is given in Table 5.

In cows treated with abomasal rolling, there was no significant difference (*p* = 0.52) in the probability of recurrence of LDA between cows with no concurrent disease and cows with concurrent diseases.

### 3.6. Laboratory Findings

The distribution of laboratory findings (on the day of admittance to the clinic) according to the outcome is given in Table 6. The *p*-values for laboratory parameters are presented for the univariate model; all *p*-values < 0.2 were considered for the multivariate model; for these, *p*-values are presented for both models. The probability of recovery increased with increased base excess (*p* < 0.05), as well as increased glutaraldehyde test, sodium, and phosphorus levels (*p* < 0.01 for each, respectively). The probability of recovery decreased significantly with increased glucose concentrations (*p* < 0.01), total protein concentration (*p* < 0.01), and concentrations of aspartate aminotransferase (AST, *p* < 0.01).

## 4. Discussion

We conducted this retrospective study to investigate the effect of either surgical or conservative treatment on recovery from LDA during time of hospitalization, to assess the influence of concurrent diseases on the recovery, and to identify prognostic indicators from an analysis of laboratory blood parameters based on the outcome in cattle diagnosed with LDA. Part of the data were evaluated with different objectives in a previous paper [13].

According to the literature, dairy breeds are more commonly diagnosed with LDA, with the predominant breeds being Holstein Frisian, Guernsey [5,6], and Jersey cows [5]. This could not be confirmed in our study. The majority of the affected cattle were German Simmental (66.4%), which is a dual-purpose breed. This can be explained by the data being collected at a clinic in Bavaria, Germany, where German Simmental is the predominant breed. The mean and SD values of days post-partum diagnosed in cattle with LDA were 13.8 ± 13.1 days, which is in accordance with other studies (mean and SD of 11.4 ± 6.8 days published by [19] and median of 15 days published by [20]). The mean and SD age in our study population was 5.1 ± 1.8 years; the highest risk for developing LDA was from 4 to 7 years, with the risk increasing with age [5].

A total of 31.1% of cattle were submitted to abomasal rolling at our clinic. The decision whether a cow was to be treated with abomasal rolling was made based on the cow’s general condition, or if claw lesions were obvious and diagnostic claw trimming was performed. Rolling only provides a short-term relief of the symptoms in cows, which cannot tolerate another procedure at that time [4], e.g., if massive electrolyte shifts or concurrent diseases are present. According to our data, LDA did reoccur in 56.7% of the cows following abomasal rolling during the time of hospitalization. This is in accordance with other data, stating that reoccurrence rates are between 50 to 70% [4]. The relapse rate might even have been higher in the cows included in this study, but as we did not track the development of the cows after discharge, we cannot make a statement about this. These results, given with the advantage of fewer expenses for this technique compared with open techniques [7] and a short procedure time [4], might make this treatment option an attractive one for farmers. Concurrent diseases might have an influence on the relapse of LDA; however, in a previous work, we found that claw lesions did not have an effect on relapse after abomasal rolling [13]. Moreover, according to the present data, there was no significant difference in the probability of the recurrence of LDA between the cows with no concurrent disease and those with concurrent diseases. LDA could not be treated with abomasal rolling in 7.2% of the cows, which is a higher number compared with 1.5% and 5.5% described by [21] and [13]. In none of the cases did the abomasum move from the left to the right side, which was described in 44.4% of cases by [21]. We did not roll the cows in a 180 degree arc but kept them in left lateral recumbency on a tilt table, whereas [21] did not keep the cows in dorsal recumbency, therefore allowing no time for the abomasum to evacuate the gas to experimentally induce right displacement of the abomasum. This could explain the differences between the studies. According to our data, analgesic and/or oral therapy did not influence relapse rates after abomasal rolling, which we already found in a smaller study population comparing differences between cows with and cows without claw lesions and LDA following abomasal rolling [13]; this is not in accordance with a previous study that found a 58% treatment success in cattle which received 50 L of an electrolyte drench following abomasal rolling [22]. As our data were obtained retrospectively, the therapy of the cows was variable. In particular, the addition of sodium sulfate and the amount of water for the oral drench depended on the surgeon, and the amount of water (10 to 40 L) administered was not always documented. The amount of water/fluid administered might have an effect on the relapse of LDA, as a higher amount of fluids might result in a “volume and weight” effect by weighting down and retaining the abomasum in its position.

Even if several methods for the correction of LDA are available, surgery offers the highest cost-to-benefit ratio [6]. The majority of the cows of our data set were submitted to surgery, with endoscopic abomasopexy and right flank laparotomy being the techniques most often used (40.8% and 40.2%, respectively). Factors influencing the chosen techniques are patient and farm circumstances [6], available sources [6], the presence of concurrent diseases [6], the direction of displacement, or the presence of adhesions of the abomasum [7]. Advantages differ between the techniques; laparoscopic abomasopexy offers the possibility of visual control and a minimally invasive method [6], with a reduction in the complications associated with right flank laparotomy [11] such as incision healing or peritonitis [4], and lower expenses for the surgery [7]. Open surgical procedures, such as right flank laparotomy, offer the advantage of allowing for direct visualization of the abomasum, as well as manual examination of the structures in the abomasum [4].

Veterinarians need to be aware of the advantages and disadvantages of the different surgical methods [6]—however, the technique chosen by the surgeon largely depends on his or her preferences [6,7], as does the method of fixation of the abomasum (according to Dirksen [15], omentopexy, or pyloropexy), which is often only decided on during surgery when the omentum and the abomasum are visible. It has been stated that a pyloropexy in combination with an omentopexy increases the strength of the pexy, as the omentum can break down or stretch, which might result in a redisplacement of the abomasum [7]. As our data were collected at a clinic, a high number of changing surgeons in different teams performed the surgeries over the years, probably influencing the distribution of treatments and surgery techniques the most, as the decision about which surgery would be performed depended on the assessment of the animal by the surgeon in charge. At our clinic, surgeries were performed by a total of 20 veterinarians over the years, resulting in different durations of surgery, which might have influenced the outcome. As the number of surgeons changed over the investigated period, the surgical teams were inconsistent and highly variable, as was the number of surgeons performing a procedure (one or two surgeons with a variable number of students), and the duration of the surgery was not recorded, these factors were not included in our statistical model—especially the factor of the surgeon could have had an influence on our results. However, in the present study, 40.8% of the surgeries were performed as endoscopic abomasopexy as described by Janowitz [16], and 76.7% of all right flank laparotomies were performed as described by Dirksen [15]. The method of the surgeries performed by the veterinarians was consistent and therefore should not have influenced our results.

It is interesting that the method of surgery did not significantly influence the days spent at the clinic. The fact of a significant difference in outcome (37% recovery and 63% death, *p* < 0.01) for cows not treated with surgery (no surgery) could be explained either by the cows’ being euthanized due to economic reasons instead of undergoing a surgical intervention or their dying prior to surgery. However, one major limitation of the present study is that, as it was performed retrospectively, we were only able to study the recovery over a very short time during the hospitalization of the cows (5 to 7 days), and not over a medium- or long-term period. The time the cows spent at the hospital could even have been too short to draw any conclusions on the incidence of relapse following abomasal rolling, as stated above. Therefore, the authors of the present study can only make a statement about recovery during the hospitalization of the animals.

A significantly higher number of cows recovered after treatment, compared with the outcome “death” (80.2% compared with 19.8%, respectively). This is in accordance with a previous study, which found that 80.7% of the cattle were discharged as cured after right flank omentopexy [23]. As the cows which were presented at our clinic with LDA were patients preselected by the referring veterinarian, and are normally not easy cases, the (long-term) recovery rates might be higher in the field practice. According to [24], 11% of cows with LDA were culled within the first 60 days, and 36% within 1 year after surgery, respectively [20]. As we did not perform a follow-up of the patients, no statement about the long-term outcome can be made. The risk of culling for cows with LDA after endoscopic abomasopexy is described to be 1.5 times higher with healthy control cows [20]. The method of surgery did not have an effect on the survival rate or days spent at the clinic. This is in accordance with a previous study published by [24] in which cattle were submitted to either left or right paralumbar or right paramedian abomasopexy. The risk of culling was 37.3 times greater for cows treated conservatively (with abomasal rolling) and 9.1 times greater for cows submitted to right omentopexy, compared with the control cows. The median intervals from calving to culling were 28 and 195 days for conservatively treated cows and cows submitted to right omentopexy, respectively [3]. To alleviate pain, pre-emptive analgesia as well as multi-modal pain management have been recommended for standing flank laparotomies in cattle [25]. The positive effects of pain management with analgesics on markers used to assess pain such as substance P [26] or heart or respiratory rate [27] have been published. As pain management was heterogenous because different surgeons performed the surgeries, resulting in a large number of analgesics being used in various combinations and often for a variable number of days, and as no parameters indicative of pain were assessed during and after surgery, an analysis of pain management and its effects on pain in cows was not conducted for the present study. Therefore, we cannot make a statement about the influence of pain management on the outcome after surgery.

Cattle diagnosed with LDA are usually presented to clinics with other conditions [12]. Ketosis, retained placenta and metritis, and hypocalcemia have been identified as risk factors for LDA [28]. According to [6], at least 50% of cattle diagnosed with LDA also suffer from a concurrent disease, of which metritis, mastitis [6,12], and ketosis are the most notable [12]; 76% of the concurrent diseases are related to the reproductive tract of the cattle [29]. In our study population, 82.4% of the cows were diagnosed with at least one concurrent disease, compared with 61.1% as described by [29]. The concurrent disease most often diagnosed in our study population was metritis/endometritis, with 38.4%; due to the retrospective nature of this work, we combined the diagnosis of retained fetal membranes, endometritis, and puerperal metritis, potentially increasing the number of cases in our study population. However, our results are within the range of previously published percentages (26% for metritis by [30], 37.3% for endometritis by [23], and 15% and 37% for retained fetal membranes and metritis by [29], respectively). Subclinical ketosis and clinical ketosis were diagnosed in 28.0% and 22.4% of our study population, respectively, which is a lower number compared with 51.9% for ketonuria found by [23], and 62% for acetonemia found by [30], respectively. However, neither the definition of ketonuria nor that of acetonemia was presented in these studies, which might explain the differences in percentages.

Hypocalcemia is stated to be an indicator for inadequate intake of feed in the prepartum period, leading to other direct risk factors for LDA, e.g., increased NEFA concentrations or subclinical ketosis [1]. The combination of the effects of hypocalcemia, ketosis, and hypokalemia indicate a multifactorial pathogenesis of LDA [31]; this combination was also given in cows in our study population. However, we did not evaluate the potassium levels, which might have influenced our results. It is also possible that cattle diagnosed with hypocalcemia are at increased risk of LDA due to decreased motility of the rumen and abomasum [28]. For the present study, hypocalcemia was defined as ionized calcium concentrations lower than 1 mmol/L according to the reference ranges used by the Clinic for Ruminants with Ambulatory and Herd Health Services, and diagnosis was performed during the admission examination at our clinic. Therefore, we do not know the number of cows which suffered from hypocalcemia at their home farm in the prepartum as well as in the postpartum period, which might have influenced the results of this study.

The percentage of the cattle diagnosed with lameness in our study population (16.7%) was within the range of the findings of other studies (from 10% [30] to 26.8% [23]). In a previous work analyzing a sub-part of the date presented in this paper, we found that 46.4% of cattle with LDA which were also submitted to a diagnostic examination of the claws were diagnosed with at least one claw lesion [13]. For the present paper, we included all cattle and not only cows submitted to a diagnostic examination of the claws into the statistical analysis, which might explain the differences in the numbers between these two works. It is possible that if we had performed a diagnostic examination of the claws in all the cows of the data set of the present study, the number for lame cattle might have been higher.

The presence of concurrent diseases did not influence the relapse of LDA following abomasal rolling, nor did the number of concurrent diseases influence the outcome. However, there was a trend (*p* = 0.08) for the number of concurrent diseases to influence the number of days spent at the clinic, with a higher number of concurrent diseases resulting in a lower number of days at the clinic. One possible explanation is that cows with a higher number of concurrent diseases were probably euthanized more often rather than receiving treatment; this could have been due to economic reasons, as treatment costs directly influence the decision for euthanasia from a farm economics perspective [32]. As we did not have any information about a long-term follow up of the patients after their discharge from the clinic due to the retrospective nature of this paper, we cannot make a statement about the influence of the nature of the concurrent diseases on the long-term survival of the cows. The authors of [24] found the most common reasons for culling cows with LDA within one year following surgery to be reproductive diseases (20%), low production (19%), and both claw and udder diseases (8.5%, respectively).

The authors of [33] investigated the hematological profile of cows diagnosed with LDA compared with sound control cows; contrary to their findings, PCV and leucocyte count as well as sodium, calcium, chloride, and glucose concentrations were not increased in our study population (cattle with LDA). However, we did not compare the findings in our study with healthy control cows, and instead relied on reference ranges—and the problem with reference ranges has been discussed elsewhere [34]. The concentrations were lower for potassium and higher for AST, which is in accordance with the previously mentioned study [33]. The authors of [23] found median PCV, leucocyte count, sodium, calcium, and potassium concentrations of 31%, 8.1 G/L, 138 mmol/L, 2.2 mmol/L, and 3.9 mmol/L in cattle diagnosed with LDA; as no reference values were presented, their data cannot be compared with those for our study population. In our study population, the probability of recovery increased significantly with increasing glutaraldehyde tests (*p* < 0.01) and decreasing concentrations of total protein (*p* > 0.01). As these two values are inflammatory markers, these results indicate that the recovery of an animal following either conservative (abomasal rolling) or surgical treatment for LDA is higher if there are no other inflammatory diseases. As for AST, the probability of recovery decreased with an increasing concentration of AST (*p* < 0.01). As increased concentrations of AST are indicative of fatty liver disease [35,36], and AST is known to be significantly higher in downer cows [36], these results are not surprising: they explain why the successful treatment of LDA in cows requires not only surgery, but also the treatment of lipomobilization, ketosis, and fatty liver disease [35]

Economic losses in cattle diagnosed with LDA are caused by, among other factors, reduced milk yield [3]. Study results about this vary. The authors of [20] found that the 305-day milk yields in cows diagnosed with LDA following endoscopic abomasopexy according to Janowitz [16] and control cows did not differ significantly. Another group of authors stated that mean daily milk production was 23.3 kg less (*p* > 0.01) in cows submitted to right flank omentopexy, 15.3 kg less (*p* > 0.01) in cows submitted to toggle suturing, and 30.1 kg less (*p* > 0.01) in cows treated conservatively, compared with control cows. By contrast, [19] showed that the milk production of a study group was significantly (*p* < 0.01) lower compared with that of a control group. Moreover, there was no difference in the cattle’s milk yield 14 and 60 days following either roll and toggle or a surgical correction of LDA [30]. For the present study, we did not evaluate the milk yields of the cows hospitalized at our clinic, as the data were inconsistent due to documentation by different caretakers. The cows received different antibiotic and analgesic treatments for a variable number of days, which might have influenced the milk yield and data. Moreover, some cows were diagnosed with LDA during their stay at the clinic following parturition or claw diseases, making the comparison of milk yield on defined days difficult. Therefore, we cannot make a statement about the development of milk yield following the different surgical procedures. Moreover, as milk yield is influenced by many factors, e.g., other diseases such as clinical mastitis [37], (puerperal) metritis [38], ketosis [39], and lameness [40], which, among others, were diagnosed as concurrent diseases in our study population, a prospective study comparing the milk yield of cattle following different surgical techniques compared with a healthy control group would be advisable.

## 5. Conclusions

The results of this retrospective study show that the method of surgery did not influence the number of days spent at the clinic for recovery, but the authors cannot make a statement about medium- or long-term recovery. Concurrent diseases such as metritis and endometritis, (subclinical) ketosis, mastitis, claw diseases, and sepsis are often present in cows with LDA and need to be diagnosed and treated alongside surgical or conservative treatment of LDA to improve the outcome of the animal. Farmers should always be informed about concurrent diseases and the necessary treatments for decision making. Conservative therapy via abomasal rolling showed a high relapse rate, indicating that farmers need to be told that this method of therapy might only result in a temporary relief, and surgery should be preferred. The possibilities of the method of surgical intervention, as well as costs, should be discussed with the farmer, even if the chosen method of surgery largely depends on the veterinary surgeon.

## Figures and Tables

**Figure 1 animals-12-01649-f001:**
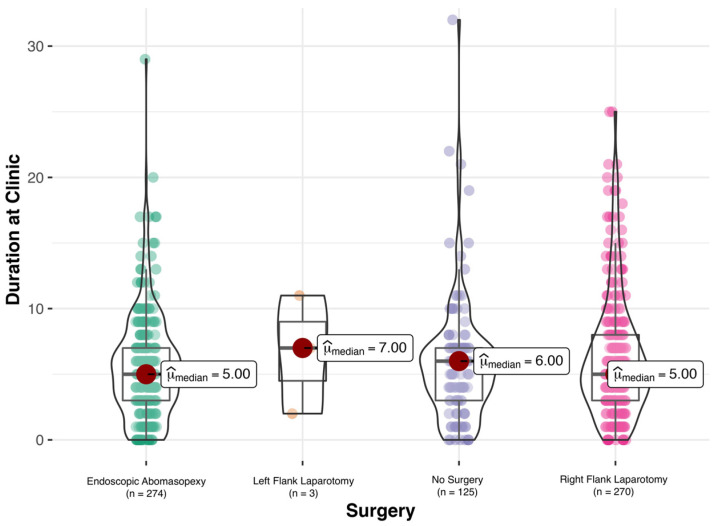
Duration of time spent at the clinic in days according to method of surgery for cows diagnosed with left displacement of the abomasum (LDA). Median numbers for days spent at the clinic were 5 days for endoscopic abomasopexy, 7 days for left flank laparotomy, 6 days for no surgery, and 5 days for right flank laparotomy. There was no significant difference for the numbers of days spent at the clinic for the different methods of surgery.

**Table 1 animals-12-01649-t001:** Distribution of analgesic and oral treatment (oral drench) in 209 cows with left displacement of the abomasum (LDA). Following abomasal rolling as conservative treatment, 79.0% (*n* = 165) and 8.1% (*n* = 17) of the cows were treated with either one or two analgesic drugs, and 45% (*n* = 95) of the cows received an oral drench. The composition of the oral drench varied and could include calcium, potassium chloride (KCl), magnesium oxide (MgO), sodium chloride (NaCl), soidumhydrogencarbonat (NaBic), sodiumhydrogenphosphate (NadPh), propylene glycole, sodium sulfate, and/or vitamin E/selenium.

Treatment	Number of Cows
Analgesic Treatment	
Dexamethason	1.4% (*n* = 3)
Flunixine meglumine	48.8% (*n* = 102)
Ketoprofen	21.1% (*n* = 44)
Meloxicam	13.9% (*n* = 29)
Metamizole	10.0% (*n* = 21)
Oral Treatment	
Calcium	22.5% (*n* = 47)
KCl	39.7% (*n* = 83)
MgO	0.5% (*n* = 1)
NaBic	(*n* = 8)
NaCl	2.4% (*n* = 5)
NadPh	2.9% (*n* = 6)
Propylene glycole	30.6% (*n* = 64)
Sodium sulfate	12.9% (*n* = 27)
Vitamin E/Selenium	37.3% (*n* = 78)

**Table 2 animals-12-01649-t002:** Manner of diagnosis of concurrent diseases in 672 cows with left displacement of the abomasum (LDA) during hospitalization at the Clinic for Ruminants with Ambulatory and Herd Health Services. Due to the retrospective nature of the study, retained fetal membranes, metritis, and endometritis were summarized as metritis/endometritis. All forms of mastitis are indicated as “mastitis”. Claw pathologies are presented as “claw disease”.

Concurrent Disease	Diagnosis
Metritis/Endometritis	- Rectal and vaginal examination - Rectal ultrasonographic examination if applicable
Mastitis	- Examination of the udder - California Mastitis test - Milk samples (sterile)
Septicemia	- Clinical examination - Laboratory analysis (leucocyte count < 4 G */L and thrombocyte count < 200 G*/L)
Subclinical Ketosis	- Clinical examination - Laboratory analysis (betahydroxybutyrate from 1.2 to 2.99 mmol/L) [18]
Clinical Ketosis	- Clinical examination - Laboratory analysis (betahydroxybutyrate > 2.99 mmol/L) [18]
Hypocalcemia	- Clinical examination - Laboratory analysis (Ionized Ca < 1 mmol/L)
Claw Disease	- Mobility scoring - Diagnostic claw trimming if applicable

* Giga.

**Table 3 animals-12-01649-t003:** Results of conservative treatment of left displacement of the abomasum (LDA) in 209 cattle (31.1% of patient population). Conservative treatment consisted of abomasal rolling by means of a tilt table at the Clinic for Ruminants with Ambulatory and Herd Health Services. In 3 cows, information about abomasal rolling (“yes” or “no”) was not given in the medical files; therefore, these were excluded from the statistical model. After abomasal rolling, LDA was solved in significantly (*p* < 0.01) more cows compared with not solved. A total of 71.8% (*n* = 150) of cows were submitted to surgery following abomasal rolling.

**Results of Abomasal Rolling in 209 cows with LDA**	**Number of Cows**
LDA solved	92.8% (*n* = 194)
LDA not solved	7.2% (*n* = 15)
**Relapse of LDA following Abomasal Rolling in 194 cows**	**Number of Cows**
Relapse: yes	56.7% (*n* = 110)
Relapse: no	43.3% (*n* = 84)
**Surgery following Abomasal Rolling in 150 cows**	**Number of Cows**
Endoscopic abomasopexy	40.0% (*n* = 60)
Right flank laparotomy	60.0% (*n* = 90)

**Table 4 animals-12-01649-t004:** Distribution of outcome according to method of treatment in 672 cows with left displacement of the abomasum. The number of cows in which surgery was aborted due to intraoperative findings is indicated. One animal died spontaneously following endoscopic abomasopexy. No surgery was performed in 18.6% (*n* = 125) of the cows, either because the cows were treated successfully with abomasal rolling or because they were euthanized or died (4 cows died on day 0 and 12 cows on days ≤ 3).

Method of Surgery	Outcome		
	Recovery	Death	Abortion of Surgery
**Right flank laparotomy**	87% (*n* = 235)	13% (*n* = 35)	2.9% (*n* = 16)
**Left flank laparotomy**	100% (*n* = 3)	0% (*n* = 0)	nA ^1^
**Endoscopic abomasopexy**	93% (*n* = 255)	7% (*n* = 19)	0.2% (*n* = 1)
**No surgery**	37% (*n* = 46)	63% (*n* = 79)	nA

^1^ Not applicable.

**Table 5 animals-12-01649-t005:** Distribution of concurrent diseases in 664 cows diagnosed with left displacement of the abomasum (LDA). Subclinical ketosis was defined as betahydroxybutyrate from 1.2 to 2.99 mmol/L [18], and clinical ketosis as betahydroxybutyrate > 2.99 mmol/L [18]. Due to the retrospective nature of the study, retained fetal membranes, metritis, and endometritis were counted as one disease, as were the different forms of mastitis (given as mastitits), and claw pathologies, indicated as claw diseases. Death or recovery were considered as survival rate. Results with a *p*-value < 0.05 were considered statistically significant and are printed in bold letters.

Concurrent Disease	Presence	Absence	Influence on Survival Rate
**Metritis/Endometritis**	38.4% (*n* = 255)	61.6% (*n* = 409)	***p* = 0.03**
Mastitis	19.6% (*n* = 130)	80.4% (*n* = 534)	*p* = 0.74
Septicemia	12.7% (*n* = 84)	87.3% (*n* = 580)	*p* = 0.48
Subclinical ketosis	28.0% (*n* = 186)	72.0% (*n* = 478)	***p* = 0.02**
Clinical ketosis	22.4% (*n* = 149)	77.6% (*n* = 515)	*p* = 0.42
Hypocalcemia	15.1% (*n* = 100)	84.9% (*n* = 564)	*p* = 0.84
Claw disease	16.7% (*n* = 111)	83.3% (*n* = 553)	*p* = 0.42

**Table 6 animals-12-01649-t006:** Distribution of laboratory findings on the day of admission in 672 cows diagnosed with left displacement of the abomasum (LDA) according to the outcome (“recovery” or “death”). Ranges and physiologic values for laboratory parameters are given in brackets; values are given as mean and standard deviation (SD). For probability of recovery, results with a *p*-value < 0.05 were considered statistically significant and are printed in bold letters. Results are given for both the univariate and the multivariate model. All parameters with a *p*-value < 0.2 were included in the multivariate model; all other parameters are indicated as not applicable (nA).

	Outcome		Statistical Model	
	Recovery	Death	Univariate	Multivariate
Laboratory Parameter			*p*	*p*
pH ^3^ (7.35–7.45)	7.4 ± 0.1 (7.22–7.64)	7.4 ± 0.1 (7.02–7.58)	***p* < 0.0006**	*p* = 0.06
Base excess ^4^ (−2.5–2.5 mmol/L)	3.0 ± 5.8 (−15.1–21.5)	1.7 ± 8.6 (-24.9–23.8)	***p* < 0.04**	nA
Packed cell volume ^5^ (30–36%)	35.2 ± 5.4 (14.3–54)	35.3 ± 6.4 (22–51.5)	*p* = 0.81	nA
Leucocyte count (4–10 G */L)	7.2 ± 3.5 (1.2–24.4)	7.5 ± 3.9 (2.1–23.2)	*p* = 0.34	nA
Thrombocyte count (200–800 G */L)	467.9 ± 191.7 (19–1261)	468.7 ± 283.8 (32–2631)	*p* = 0.97	nA
Anion gap ^6^ (14–26 mEqu/L)	16.1 ± 6.2 (2.7–106)	16.9 ± 6.7 (2.3–41.3)	*p* = 0.21	nA
Glucose ^7^ (2.5–3.3 mmol/L)	5.0 ± 2.1 (1.4–16.7)	5.5 ± 3.2 (1.8–33.6)	***p* < 0.03**	nA
L-lactate ^8^ (≤2.2 mmol/L)	2.3 ± 2.1 (0.26–14.22)	3.1 ± 2.8 (0.15–14.29)	***p* < 0.001**	*p* = 0.27
Total protein (60–80 g/L)	75.3 ± 9.5 (34.9–116.1)	77.9 ± 12.7 (48.3–126.3)	***p* < 0.009**	nA
Aspartate aminotransferase ^1^ (≤80 U/L)	286.2 ± 305.3 (29.1–5395.2)	401.0 ± 510.1 (10.1–2918.8)	***p* < 0.003**	***p* < 0.01**
Gamma glutamyl transferase ^2^ (≤36 U/L)	52.0 ± 67.8 (6.8–624.7)	61.1 ± 91.9 (5.4–697.5)	*p* = 0.02	nA
Glutamate dehydrogenase ^3^ (≤16 U/L)	83.4 ± 136.7 (2.32–1261.26)	123.3 ± 348.9 (1.92–3329.48)	*p* = 0.06	nA
Betahydroxybutyrate ^9^ (≤1.0 mmol/L)	2.0 ± 1.8 (0.08–10.14)	2.0 ± 2.9 (0.1–14.96)	*p* = 0.83	nA
Glutaraldehyde test ^10^ (<15 min)	10.1 ± 5.8 (0.5–16)	7.3 ± 6.2 (0.5–16)	***p* = 1.383 × 10^−6^**	***p* < 0.01**
Sodium ^11^ (135–150 mmol/L)	138.1 ± 3.8 (117–157.4)	135.6 ± 4.7 (119–145.9)	***p* = 6.077 × 10^−10^**	***p* < 0.01**
Potassium ^12^ (4–5 mmol/L)	3.3 ± 0.5 (1.71–4.91)	3.1 ± 0.7 (1.86–4.86)	***p* < 0.002**	nA
Ionized calcium ^13^ (1.0–1.3 mmol/L)	1.1 ± 0.6 (0.64–1.53)	1.1 ± 0.1 (0.71–1.99)	*p* = 0.11	*p* = 0.08
Chloride ^14^ (90–105 mmol/L)	98.1 ± 6.3 (65–115)	95.5 ± 8.2 (67–110)	***p* < 0.0001**	*p* = 0.24
Phosphorus (1.5–2.1 mmol/L)	1.5 ± 0.6 (0.2–4.1)	1.7 ± 0.7 (0.3–3.9)	***p* < 0.002**	nA
Magnesium ^15^ (0.74–1.44 mmol/L)	0.8 ± 0.2 (0.34–1.7)	0.9 ± 0.3 (0.42–3.28)	*p* = 0.52	nA

* Giga ^1^ Values missing for *n* = 1 in recovery; ^2^ values missing for *n* = 78 in recovery and *n* = 21 in death; ^3^ values missing for *n* = 9 in recovery; ^4^ values missing *n* = 5 for recovery; ^5^ values missing for 1 in recovery; ^6^ values missing for *n* = 10 in recovery and *n* = 2 in death; ^7^ values missing for *n* = 1 in recovery; ^8^ values missing for *n* = 1 in recovery; ^9^ values missing for *n* = 5 in recovery; ^10^ values missing for *n* = 19 in recovery and *n* = 1 in death; ^11^ values missing for *n* = 3 in recovery; ^12^ values missing for *n* = 5 in recovery and *n* = 1 in death; ^13^ values missing for *n* = 6 in recovery and *n* = 2 in death; ^14^ values missing for *n* = 3 in recovery and *n* = 1 in death; ^15^ values missing for *n* = 4 in recovery.

## Data Availability

The data presented in this study are available upon request from the corresponding author. The data are not publicly available due to the protection of the data privacy of the patients of the Clinic for Ruminants with Ambulatory and Herd Health Services, LMU Munich, Germany.

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
