# Peer review of "Retrospective Evaluation of Method of Treatment, Laboratory Findings, and Concurrent Diseases in Dairy Cattle Diagnosed with Left Displacement of the Abomasum during Time of Hospitalization"

_animals, 2022, doi:10.3390/ani12131649_

Round 1

Reviewer 1 Report

This paper presents very interesting data relating to the treatment of left displaced abomasum in dairy cattle over a 10 year period. It is a companion paper to an article published in Animals in 2021.

Major revisions are required.

Overall issues

One major issue with this paper is that the aims are not clear or consistent across the paper- ie:

Abstract -aims

For the present study, the medical data of 718 cattle with LDA were analyzed retrospectively to 26 assess surgical treatment, laboratory findings, concurrent diseases, and outcome.”

Introduction- aims

The objective of the present study was to retrospectively evaluate 1) the treatment method used for correction of LDA, 2) the outcome after conservative or surgical treatment, 3) the findings of blood analysis, and 4) the number and character of concurrent diseases for patients of the Clinic for Ruminants with Ambulatory and Herd Health Services in Bavaria, Ludwig Maximilians University (LMU) Munich, Germany.”

Discussion- aims

We conducted this retrospective study to investigate the distribution of treatment (either conservative or surgical) of cattle diagnosed with LDA at our clinlic, and to evaluate blood parameters, concurrent diseases, and outcome after abomasal rolling with or without analgesic and/or oral therapy.”

Clearer aims, which would map to testable hypotheses might be as follows

The aims of this retrospective study were to

1) Investigate the effect of treatment method on recovery from LDA

2) Assess if recovery from non-surgical treatment of LDA was influenced by concurrent disease

3) Identify prognostic indicators from laboratory analysis of blood samples, based on subsequent outcomes

Presentation and analyses of results

This lack of clear aims also leads to a confusing presentation of the results. Although substantial statistical methods appear to have been applied, what is unclear is the specific comparisons which have been made, or exactly what hypothesis is being tested with every data set and applied statistics. This is particularly an issue with the data presented in figures 1 and 3.

Ideally, the layout of the methods, results and discussion should be consistent with stated aims and follow a matching pattern. The discussion focuses initially and excessively on breed and post-partum data, yet this is not a stated aim of the paper.

Sequencing of the material is not consistent or logically presented- for example Biochemical parameters are at 2.3 in the methods, presented in section 3.5 (outcome) in the results, and are the second section within the discussion.

Specific points

Lines: 

64-67: Clearer objectives needed

73: Need some more detail regarding "admittance".

Were these "first opinion" or referred by "outside" veterinary clinicians? This needs to be addressed from the outset, and not only in discussion as affects the determination of perceived difficulty of the LDA cases.

Provide little more detail on the Clinic for Ruminants in Munich, and the area it serves.

100: Was this rolling with "tacking", or fully conservative?

101-102: The Dirksen and Janowitz papers are in German and not generally available, so could you please give more detail, or refer to papers that the broader audience can check- as the exact techniques, including the degree to which the abdomen is opened, whether a stitch/ toggle is put in place etc is not clear here.

Suggest Niehaus (2016) Vet Clin Food Anim 32 (2016) 629–644 http://dx.doi.org/10.1016/j.cvfa.2016.05.006

177: Are these results only for those animals which were rolled? 

I am not seeing overt statistics to support the statement made regarding the influence of concurrent disease. Statistical comparisons appear to be within and not between "concurrence no." How does having no concurrent disease have a significant influence on relapse? Yes, the number relapsing is larger than the number non-relapsing, but that doesn't mean the lack of concurrent disease is "causal". "No concurrent disease" should be the "control" group for a question on whether the presence of concurrent disease increases the chance of recurrence. 

193-194. These results are out of place and should appear before line 176.

Figure 2:

This figure appears incomplete. Is the y axis "number of animals"? However, this figure serves little purpose, and a table of medical treatments will suffice. It may be better to present these treatments under "methods". There is little to distinguish between the different NSAIDS, nor would I expect these oral treatments to have any effect.

212: What is meant by the "abomasum being in situ"? Again, the precise surgical techniques, and distinctions between them need to be better described in the methods section.

Figure 3: What was the hypothesis or specific research question which drove the statistical analysis to compare choice of surgery within years?

Why compare these, and what useful information does this offer? The aims state that these were evaluated, but a "count" is not an evaluation.

Figure 4. Use the full term "endoscopic abomasopexy" throughout the paper.

255: It may be useful to reiterate that these were "day of admission" laboratory results.

Table 2: States that age and post-partum data is included- it is not.

Figure 5: Again, these p values appear to result from within treatment comparisons, not between treatment comparisons. What was the hypothesis that drove this comparison choice? The p values on the image are not helpful.

A useful, testable (True vs False) hypothesis would be :  "Non-surgical intervention is more likely to result in death, in comparison with right flank laparotomy and endoscopic abomasopexy"

If these comparisons were made, please report them more clearly.

Lines 289-292: Aims do not match the previously stated aims. I did not see a comparison provided in the results for outcomes with or without analgesia.

353: I was under the impression that the presented results were the “rolled” and surgical animals combined- if so, this statement refers to results for surgery only that were not provided.

443. That the animals were referred/ second opinion needs to be in the introduction, and materials and methods.

457: The data has not been clearly presented to support this statement.

Some typos and further comments are shown within the attached file.

Reviewer 3 Report

Please,

See file attached.

Reviewer 4 Report

This paper analyzes a retrospective study of LDA with a significant number of clinical cases with relevant conclusions to the readers.

Minor errors in the edition are necessary to correct:

line 443: separate the words

line 523: correct the cite

line 539  treatment

Round 2

Reviewer 1 Report

Dear authors,

I appreciate the substantial changes that have been made in response to reviewer comments.

Please check the paper again for typographical errors. I found quite a few, but probably not all. An annotated file is attached.

Reviewer 3 Report

The manuscript has improved markedly in the amount of information provided and in the structure. Although some spelling mistakes are present (please see below).

There is one main concern that should be very well explained throughout the paper:

The authors need to be clear about the limitations of the study. Especially the most important: This study only observes and comes to conclusions about a very short recovery period, which is the stay in the clinic. But it does not study the recovery in the medium and long term. In my opinion, the authors should explain this throughout the manuscript, especially in the title, abstract, discussion (they already do) and conclusions. The time period studied is even too short to conclude on the incidence of recurrence in the "Rolling the cow" treatment...

Line 153: veterinarians

Line 169: Please, check and rewrite.

Line 277: treatment

Line 377: abomasopexy

Line 555: of

Line 611: direct

Line 633: abomasal
